# Context-Guided Medical Visual Question Answering

No Institute Given

**Abstract.** Given a medical image and a question in natural language, medical VQA systems are required to predict clinically relevant answers. Integrating information from visual and textual modalities requires complex fusion techniques due to the semantic gap between images and text, as well as the diversity of medical question types. To address this challenge, we propose aligning image and text features in VQA models by using text from medical reports to provide additional context during training. Specifically, we introduce a transformer-based alignment module that learns to align the image with the textual context, thereby incorporating supplementary medical features that can enhance the VQA model's predictive capabilities. During the inference stage, VQA operates robustly without requiring any medical report. Our experiments on the Rad-Restruct dataset demonstrate a significant impact of the proposed strategy and show promising improvements, positioning our approach as competitive with state-of-the-art methods in this task.

**Keywords:** Medical Visual Question Answering · VQA · Medical Image Interpretation · Radiology

## 1 Introduction

Medical Visual Question Answering systems can streamline healthcare workflow efficiency by allowing quick retrieval of relevant information from medical images. This can save valuable time for healthcare providers and offer additional insights into diagnostic procedures while assisting in clinical decision-making. Despite holding such potential, the application in the medical field has been minimal due to the small-scale of available datasets, the complex nature of medical images, the diversity of questions and the high level reasoning required to answer them.

Initial research [5,11] attempted to transfer advances in general VQA to the medical domain. However, Medical VQA faces unique constraints related to the acquisition and processing of data [16]. Usually, constructing medical VQA datasets require costly expert annotation and professional knowledge, *e.g.*, extracting Question-Answer (QA) pairs directly from a medical image needs domain specific expertise. This limitation often restricts the data collection process leading to small size datasets. Unlike general-domain VQA datasets, such as VQA [3], that contains hundreds of thousands of samples, medical VQA datasets are limited to tens of thousand images and QA pairs[4,6,7,14].

Recent research focuses on harnessing the potential of attention-based pre-trained biomedical visual language models such as BioGPT [18] and BLIP-2[15] in a generative strategy. For instance, Van Sonsbeek *et al.* [24] proposed mapping visual features to a set of tokens that prompt a GPT-XL decoder [23]. For classification-based VQA, capturing high-quality medical features is crucial. For this purpose, Mixture Enhanced Visual Features model (MEVF) [19] uses the Denoising Auto-Encoder (DAE) as a visual extractor to enhance the quality of visual features extracted from medical images. With a typical focus on either visual feature extraction or fusion models, recent works [12,2] have seen a surge in the deployment of vision-language models as image encoders for their significant capability of producing high-quality features. However, such approaches rely heavily on transformers and attention mechanism for feature aggregation and fusion [25]. While attention has proven to be effective in cross-modal settings [2] and text-image fusion [1], this ties the advancement of medical VQA to the development status of attention mechanism and restricts the progress to fusion models. In contrast, this work posits that incorporating additional information to provide textual context during training could enhance both feature extraction and feature fusion.

Inspired by the NLP Question Answering task [20], we propose to use free-text medical reports as additional context to enhance visual feature extraction during training. Specifically, medical reports provide context we use to align image features with the same embedding space as the questions. This alignment improves the quality of image features, enabling accurate responses to questions during the inference stage without requiring additional context input. Our focus is on a closed-ended classification VQA task, where the targets are a predefined set of answers. In our pipeline, we first summarize medical reports using the GPT model [8] to effectively clean and pre-process the raw free-text for better encoding. Then, our model incorporates context features from summarized medical reports through a trainable alignment module that learns meaningful correlations between the textual and visual features. This enhances the image content as a reference for the question and helps identify fine-grained visual details guided by the medical text. To generate the final classification input of the VQA model, a fusion module is trained to combine the aligned image-context features and the encoded question features. Moreover, we use a pre-trained visual language encoder as an image encoder and an LLM-based encoder for text encoding without extra finetuning. By doing so, we significantly reduce the number of parameters and the computational complexity of our model. Experiments on the Rad-Restruct dataset [21] show an increase in performance outperforming the baseline model, with a state-of-the-art accuracy.

In summary, our research contributes in three distinct aspects. First, we propose the first approach to use free-text medical reports as context to guide the prediction of answers in medical VQA. Second, we harness the power of pre-trained models throughout our model while reducing computational complexity by 90%. Finally, we prove through experiments that our strategy outperforms the baseline methods on different metrics with state-of-the-art accuracy.

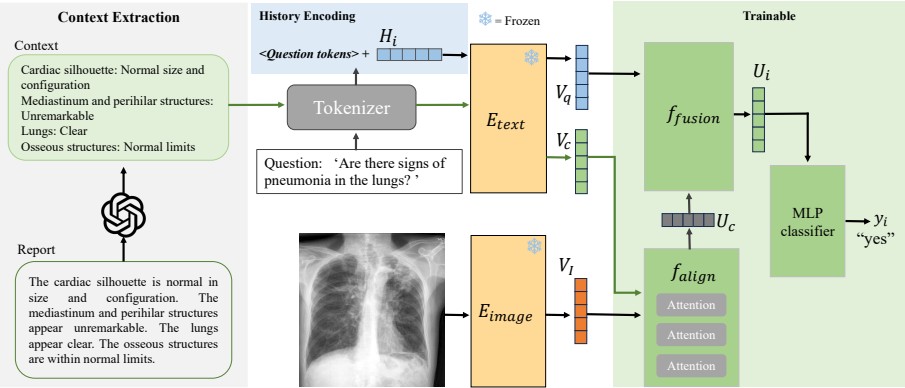

**Fig. 1.** Model architecture of our proposed context-guided VQA method. $E_{text}$ and $E_{image}$ denote, respectively, an image and a text encoder . $f_{align}$ is an alignment network and $f_{fusion}$ denotes a fusion model. The process of extracting the context from free-text reports is illustrated on the leftmost side. The encoders extract $V_q$, $V_c$, and $V_I$, then the context embedding $U_c$ is aligned in $f_{align}$. Next, $f_{fusion}$ produces a final embedding vector $U_i$ which serves as the input of the MLP classifier that predicts the answer $y_i$. The history vector $H_i$ contains tokens of all the previous questions with their answers.

## 2    Methodology

Figure 1 presents an overview of our architecture. The primary objective of this method is to utilize free-text reports to enhance the visual reference space, thereby providing a more comprehensive context for answering questions. To achieve this, we first encode the medical report associated with each image to generate textual context, and introduce a module to align this context with image features. Subsequently, we fuse the context-image features with the question vector and feed them into an MLP classifier for final answer prediction. Our framework has 5 main components: 1) a text encoder for encoding the question and context, 2) an image encoder for encoding image features, 3) an alignment module that learns to align and fuse the context and image features, 4) a fusion model that learns to combine the aligned context-image vector with the question features, and 5) a classifier that is trained to predict the answer.

### 2.1    Report Context Processing

The majority of free-text medical reports are divided into sections *i.e.*, *Impression* and *Findings*. Following prior literature on report generation [13], we focus on the *Impression* section as it provides less general information, and more detailed medical findings. The extracted text is summarized using a prompted GPT model and the output summary is used as context. The summarizing process aims not only to potentially shorten the reports but also to clean and

simplify the text descriptions. Herein, we denote the training data as follows: $D = (c_i, q_i, I_i, y_i)_{i=1}^{N}$ where $c_i$ represents the context (text), $q_i$ the question (text), $I_i$ is the image, and $y_i$ the corresponding ground truth label (text).

## 2.2 Model Architecture

For an input image $I_i$, we extract the visual features $V_I$ using a pre-trained image encoder $E_{image}$. Formally,

$$V_I = E_{image}(I_i). \tag{1}$$

Similarly, we obtain feature vectors of the question $V_q$ and context $V_c$ through a text encoder $E_{text}$ as follows:

$$V_q = E_{text}(q_i), \quad V_c = E_{text}(c_i). \tag{2}$$

To learn correlations between the image and context, we integrate a transformer-based alignment network $f_{align}$ that produces a final context embedding $U_c$ as follows:

$$U_c = f_{align}(V_c, V_I). \tag{3}$$

$f_{align}$ is a stack of three multi-head self-attention layers that compute cross-attention scores between textual and image features. Here, we consider context as query, and image embeddings as key-value pairs. Each layer learns to attend to both image and textual context, producing context-aligned image features. The final aligned representation is obtained via the upper attention layer. The aligned image features implicitly represents image regions that are most relevant to the textual content.

Following the work of Pellegrini *et al.* [21], we adopt an autoregressive strategy and incorporate history information as input. At each step, the history vector $H_i$ contains the previous (higher-level) questions with their answers. For example, the question *"In which part of the body?"* requires prior knowledge from higher level questions such as *"Is there an opacity in the lung?"*. The history vector is constructed by concatenating previous and current question tokens.

Next, a multi-modal transformer-based fusion model $f_{fusion}$ is trained to combine the question and the image-context features to generate a final embedding vector $U_i$ for classification. Formally,

$$U_i = f_{fusion}(U_c, H_i, V_q), \tag{4}$$

where $H_i$ is the history information for the current question $q_i$. Finally, the MLP classifier predicts the answer $y_i$ given the features of the resulting $U_i$.

During the training phase, the alignment module is trained using both the context and the image features to produce a vision-context vector. Similar to [21], we train our model with a cross-entropy loss in an autoregressive manner taking in consideration the question history. For inference, our approach assumes an absence of context and takes as input the image and questions. The alignment module creates a context vector using the image through the learned textual representation during the training process.

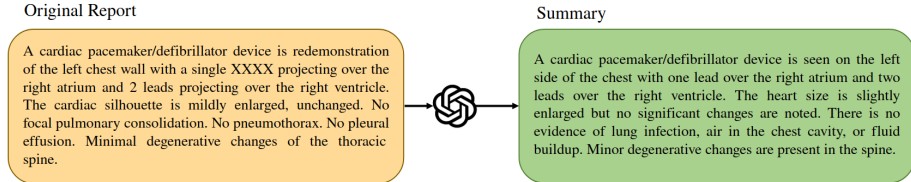

Original Report

A cardiac pacemaker/defibrillator device is redemonstration of the left chest wall with a single XXXX projecting over the right atrium and 2 leads projecting over the right ventricle. The cardiac silhouette is mildly enlarged, unchanged. No focal pulmonary consolidation. No pneumothorax. No pleural effusion. Minimal degenerative changes of the thoracic spine.

Summary

A cardiac pacemaker/defibrillator device is seen on the left side of the chest with one lead over the right atrium and two leads over the right ventricle. The heart size is slightly enlarged but no significant changes are noted. There is no evidence of lung infection, air in the chest cavity, or fluid buildup. Minor degenerative changes are present in the spine.

**Fig. 2.** An example of a summarized report by the GPT model. The generated summary has been cleaned of special characters, privacy tokens and measurement numbers, and simplified to use more natural language.

## 3 Experiments

**Dataset.** We use the Rad-Restruct dataset [21], which contains 3720 Xray images, 3597 structured reports, and 180k questions. For each image, a corresponding free-text report is retrieved from the IU-Xray dataset [9]. Images were normalized and cropped to 224×224. We use the original 80-10-10 split for training, validation, and testing, respectively. Rad-Restruct questions are categorized into 3 levels. Level 1, the highest level, contains general-purpose questions such as *"Are there any foreign objects?"*. Level 2 questions ask for more specific findings, like *"Is there pneumonia in the lung?"*, and level 3 questions are related to detailed findings, for example, asking for the degree or a description of a disease, *"What are the attributes?"*. This hierarchical structure allows an autoregressive parsing of the questions, making lower-level questions depend on higher-level ones, which aims to provides more context for difficult questions in level 3 question. In addition, there are 96 classes in total. The questions can be single-choice or multiple-choice, with each having a defined set of possible options. To the best of our knowledge, at the time of the experiment, Rad-Restruct is the only available dataset that enables our experimental design by providing access to all three components, medical images, free-text reports, and QA pairs.

**Context Extraction.** To extract the context from radiology reports, we use the prompted GPT 3.5 Turbo with the following prompt: *'Please summarize the following X-ray report while keeping the medical terms.'* Figure 2 shows an example of a summarized report by the GPT API. The GPT model aims to generate natural language text comprehensively. In doing so, it rephrases reports into simplified vocabulary, excluding sensitive medical details that could be beneficial. We address this concern by requesting the retention of medical terms. The length of generated summaries varies based on the original report and content nature. We impose a maximum size of 512 tokens, truncating when needed. These summaries serve as a direct context for training our model.

**Training and Evaluation.** As an image encoder, we use PubMedClip [10], a variant of the CLIP model [22] designed for medical VQA, it is pre-trained

**Table 1.** Performance comparison on Rad-Restruct dataset. We compare 5 methods, a visual baseline [21], hi-VQA, both the published results and the released dataset results, re-VQA (our reproduced hi-VQA results) and our method Context-VQA.

|  | Report Accuracy | F-1 | Precision | Recall |
|---|---|---|---|---|
| Visual baseline [21] | 31.3 | 30.7 | 65.6 | 31.2 |
| hi-VQA [21] | 32.6 | 31.7 | 70.7 | 32.1 |
| hi-VQA [1] | 30.2 | **31.9** | 64.6 | 33.3 |
| re-VQA | 29.8 | 28.7 | 61.2 | 29.8 |
| Context-VQA | **39.7** | 31.0 | **90.4** | **33.6** |

on several medical image modalities. Following the model introduced in [21], we adopt the RadBert model as a text encoder, with the Bert Tokenizer. RadBert is a domain-specific large language model based on Roberta model [17], and was trained on a large corpus of radiology reports. To encode text inputs, we leverage RadBert's pre-trained embeddings, which capture domain-specific semantics and contextual. The encoders are frozen, preserving their pre-trained weights and preventing further parameter modification. As a classifier, we use a 5-layer MLP with batch normalization and a dropout rate of 0.2. The model is trained for 200 epochs on a NVIDIA RTX A6000 GPU. We use Adam Optimizer with a learning rate of 1e-5 and a batch size of 32.

## 4 Results

Table 1 shows the performance of our Context-VQA method and the state-of-the-art method hi-VQA [21] on the Rad-Restruct dataset. For a fair comparison, we use the same evaluation approach as hi-VQA and report our results accordingly using the conventional metrics Accuracy, F1, Precision, and Recall. As accuracy, we provide the report accuracy, a metric that is specific to the dataset.

Rad-Restruct was built to structure the text reports in the IU-Xray dataset into a form populated by hierarchical questions. Thereby, each image is accompanied by its set of questions referred to as a structured report. The report accuracy metric represents the accuracy of reports that were fully predicted correctly. Thus, a report is considered correct if all questions at all levels for a given image are perfectly answered. Context-VQA outperforms hi-VQA on this specific metric with almost a 10% increase. This translates to an increase of fully predicted reports' questions by 10%. The evaluation was done using the script provided in [21], allowing direct comparison to hi-VQA. Our results significantly outperform the hi-VQA on the other metrics, with precision improved by 29%.

Furthermore, Table 2 shows the performance on each level of questions. Context-VQA steadily achieves state-of-the-art accuracy on all levels. Although, the F-1 score shows a slight decrease for some type of questions in level 2 and level 3, level 1 and level 2 (all) show higher overall performance. In addition,

---

[1] Released dataset results https://github.com/ChantalMP/Rad-ReStruct/tree/master

**Table 2.** A comparison of the baseline Hi-VQA and Context-VQA for each question level. Context-VQA scores the best accuracy over all levels, while F1 and Recall alternate between the two models on different levels.

|                  | hi-VQA   |      |           |        | Context-VQA |      |           |        |
|------------------|----------|------|-----------|--------|-------------|------|-----------|--------|
|                  | Accuracy | F-1  | Precision | Recall | Accuracy    | F-1  | Precision | Recall |
| Level 1          | 33.6     | 64.3 | 81.0      | 64.5   | **34.7**    | **67.2** | 80.7  | 61.2   |
| Level 2 (L2) all | 31.0     | 71.6 | 85.2      | 72.0   | **32.9**    | **71.8** | **88.9** | 70.8 |
| - L2 diseases    | 48.1     | 73.5 | 83.8      | 71.3   | **52.1**    | 72.8 | **89.6**  | **72.7** |
| - L2 signs       | 71.9     | 74.2 | 93.1      | 74.4   | **74.3**    | 73.7 | 90.6      | 73.7   |
| - L2 objects     | 87.4     | 67.0 | 77.1      | 67.5   | **91.4**    | **67.2** | 85.0  | **68.6** |
| - L2 regions     | 52.4     | 68.1 | 82.1      | 69.5   | **61.2**    | **68.7** | 85.4  | 68.3   |
| Level 3          | 30.2     | 4.1  | 49.9      | 6.2    | **32.5**    | 3.2  | **68.7**  | 4.2    |

**Table 3.** The computational complexity of our leveraged model against the full architecture and the baseline

|                          | hi-VQA  | Context-VQA | Fully trained Context-VQA |
|--------------------------|---------|-------------|---------------------------|
| Number of parameters (M) | 164     | 16          | 230                       |
| Training time per epoch  | 1h30min | 35min       | 1h48min                   |

it is noteworthy that our approach, with only 16M trainable parameters was able to perform closely or even better than hi-VQA (164M parameters). We also note that we compare to the released dataset results instead of the scores reported in the paper. Figure 3 illustrates qualitative examples of predictions of Context-VQA compared to the baseline predictions. Questions are arranged sequentially from left to right, level-wise, indicating their hierarchical dependency. In the first instance, hi-VQA predicts a negative response to the initial question, consequently influencing subsequent questions to also receive negative predictions. In examples 2 and 3, we observe that for these cases Context-VQA accurately predicts lower-level questions, which are often challenging and influenced by preceding questions. This explains the ability of our model to correctly predict sequential questions leading to the improvement in report accuracy, as previously reported.

**Computational complexity**  By leveraging the pretrained encoders, we achieve a substantial reduction in the number of trainable parameters,from 230M to 16M, against hi-VQA's 164M parameters. Consequently, the training time is reduced. Experiments on a single A6000 GPU record the training time of 1h30min per epoch for hi-VQA against 35min required by Context-VQA, as demonstrated in Table 3. Notably with fewer parameters and less computational time, Context-VQA still performs well on the Rad-Restruct dataset.

**Ablation Studies**  We conducted ablation studies to assess the impact of contextual information on the prediction capabilities of our Context-VQA model. In the first experiment, we integrated the context as an input in the baseline

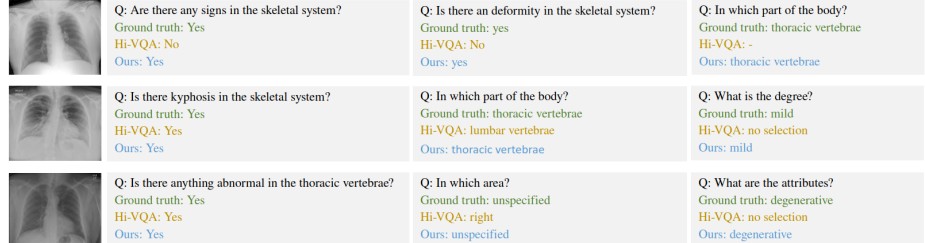

**Fig. 3.** Examples of the predictions of Context-VQA and the hi-VQA model. The questions are successive from left to right, representing the hierarchical dependency.

**Table 4.** Comparison experiment investigating the impact of the context and the alignment module.

|  | Report Accuracy | F-1 | Precision | Recall |
|---|---|---|---|---|
| hi-VQA + context | 32.3 | 30.4 | 76.6 | 30.2 |
| Ours $-f_{align}$ | 32.6 | 28.7 | 80.0 | 28.8 |
| Context-VQA | **39.7** | **31.0** | **90.4** | **33.6** |

hi-VQA model by tokenizing and feeding it into the text encoder alongside the question. The textual features were then concatenated with the visual features and processed in the fusion module. This experiment aimed to demonstrate the impact of incorporating textual information. The results, labeled as "hi-VQA + context," are presented in Table 3, showing a performance improvement compared to the baseline model. To further examine the significance of the alignment module, we adopted a similar approach of incorporating contextual information as an additional input in our model without further alignment. The results are reported as "our $-f_{align}$" in Table 4. We note that the recall and F1 scores exhibited a slight decrease, primarily attributed to the lack of encoder updates, which limited improvements in feature extraction.

## 5 Conclusion

In this work, we proposed a novel approach to enhance medical Visual Question Answering (VQA) systems by leveraging free-text medical reports as contextual information. We use the context by incorporating an adaptive text-image alignment module that learns to align textual and visual features. Through extensive evaluation on Rad-Restruct dataset, we validated the efficacy of integrating context-based information alongside images and questions that provides richer medical features for VQA with significant performance gains. Given the rise of datasets with additional medical reports and EHR, this work paves the way for further exploration and refinement of context-based approaches in advancing the capabilities of medical VQA systems and ultimately improving medical AI systems' outcomes.

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
