# OpenReview forum: "Context-Guided Medical Visual Question Answering"
_MICCAI.org/2024/Workshop/MSB — MICCAI Student Board EMERGE Workshop 2024 Oral_

### Official Review · Reviewer_Lh8y · 2024-07-05

**Recommendation:** 1
**Confidence:** 5

**Clarity:**

The paper has significant clarity issues that hinder understanding, substantial revision is required to improve clarity

**Feedback:**

There are medical errors during the technical development process by inspecting the example in figure 1.
In figure 1 context extraction, there is no legend to indicate the meaning of the symbol from report to context.
Also the content of the data contains error. The report shows the lungs are clear, which can be seen from the image. But the VQA says that there are signs of pneumonia in the lungs, which contradicts with the report and image showing that the lungs are clear.

It is unclear how accurate is the step of 2.1 report context process. For example, in figure 1, the report says "The osseous structures are within normal limits", but the extracted context is "Osseous structures: normal limits". From the context text, it is unclear whether they are "on" or "within" the normal limits, which may alter the meaning of the original report. Therefore, the authors should assess how accurate is the step of 2.1 report context process. If the model learns from imprecise medical contexts, this may not be reflected in  the task performance, but can impact how accurate the model communicates with clinicians.


It is unclear what attributes mean in this context: "for example, asking for the degree or a description of a disease,
"What are the attributes?"."

In figure 2, why the measurement numbers are cleaned from the summary? Measurement information can be important information to answer questions and to clinical judgment.

"We also note that we compare to the released dataset results instead of the scores reported in the paper."
Please clarify this sentence, what released dataset results and in which paper were the authors referring to?

In the dataset section, it is unclear whether the questions in level 1&2 are mainly yes/no questions, and the questions in level 3 are mainly multiple choice questions. And how the performance metrics are calculated? In each level, it is unclear how many questions in total are used in the performance calculation, and how many are yes/no and multiple choice questions.

**Justification:**

This work proposes techniques based on what is available of the dataset and the main trends of task formulation in AI, rather than what is really needed in clinical settings. This is not good research as it only follows the technical trends but lacks critical thinking of the common practice in the technical community and ignores the true research problems. I would like to kindly remind the authors that technical trends can easily be misleading and cannot always justify themselves. For example, "Historically grown practices are not always justified" (Understanding metric-related pitfalls in image analysis validation https://www.nature.com/articles/s41592-023-02150-0) I encourage the authors to exercise their critical thinking skills in the problem formulation and justification, and conduct good future research.

**Reproducibility:**

Not enough amount of details available for reproducing the main results, and open access details are unclear

**Strengths:**

The methodology and model description are clear.

**Summary:**

This work proposes a technique on the problem of closed-ended medical VQA by utilizing medical report and image-text alignment.

**Weaknesses:**

The problem formulation of the closed-ended medical VQA and the proposed technique exhibits vague clinical values, utilities, and benefits. It is unclear how it can contribute to the usage in clinical settings, other than the narrow motivation to "show promising improvements" and "competitive with state-of-the-art methods in this task." The authors do not provide justifications that improving performance of the closed-ended medical VQA problem is on the right track to clinical usage. If the whole research direction of the closed-ended medical VQA problem is not on the right track towards clinical utility, how can the authors justify that such performance improvement on this wrong track is valid and exhibits research significance?

It is difficult to see that the formulated problem of the closed-ended medical VQA is on the right research track. Think about the clinicians using the model, how can the model be useful to them if it can only answer questions based on the predefined n sets of multiple choice options, where the clinicians questions can be any questions? If such closed-ended answers are useful for clinicians somehow, since it's closed ended, the ultimate information the model outputs is similar to a simple classifier outputting the n sets of answers. If so, how do the authors justify the clinical utility of the proposed closed ended model, compared to a simple classifier outputting the same sets of answers without clinicians' efforts to ask questions, something like the simple classifier outputting a structured medical report? Wouldn't that simple classifier be more useful for clinicians than the proposed closed ended VQA?

If the authors assume that closed ended VQA can be the prerequisite for open-ended VQA, they did not justify this assumption in the paper. In fact, it is hard to be convinced that open and close ended VQA are the same problem, rather than two distinct ones that require different assumptions and modelling approaches.

---

> ### Author Response · Authors · 2024-07-14
> **Rebuttal by Authors**
>
> We thank the reviewer for the thorough review and valuable feedback. We greatly appreciate the time and effort they have invested in evaluating our work. We would like to address some of the concerns raised and provide clarifications on several points to highlight the contribution and clinical relevance of our research.
>
> Closed-ended medical VQA: In clinical usage, while closed-ended questions may seem limited, they offer a controlled and consistent way to validate and integrate AI in clinical settings. This ensures that the answers are reliable and align with predefined clinical standards, which is crucial for building a trustworthy VQA. In contrast, open-ended questions offer flexibility but require robust reasoning. Striking the right balance between these question types is essential for optimizing clinical utility. Although the primary focus of our work does not center on assessing the validity of closed-ended medical VQA in relation to its open-ended counterpart, we believe the formulation of the closed-ended VQA problem is a steppingstone towards more complex and open-ended systems. Our motivation lies in overcoming challenges such as data sparsity and leveraging available information to enhance VQA, ultimately bridging the gap between research and clinical practice.
>
> Clinical Utility: In both open-ended and closed-ended tasks, the set of questions is predefined within the datasets. The key distinction lies in the nature of the answers: free-text or a set of candidate answers. Notably, this difference would not pose a significant issue in terms of the clinicians’ questions. Specifically, within practical medical contexts, questions posed by clinicians are limited by the image and test type and have a finite range of potential answers. Thus, close-ended questions can be as relevant as open-ended ones.
>
> Vision Classifiers: While a simple visual classification may yield predictions, it remains confined to providing a general observation for specific diseases. In contrast, medical VQA represents a superior multimodal task. By jointly combining both images and questions, it extends its predictive capacity to encompass a broader spectrum of potential answers.
>
> Our model enables sequential question answering. If deployed as a clinical system, users would only need the image and questions without the need for context. In a real scenario, a radiologist may initially ask, "Is there an abnormality in the image?" The system responds affirmatively and supports subsequent detailed inquiries such as "What is the nature of the abnormality?", then "What are the specific signs observed?", "Where is the abnormality located?", and "What are its characteristics (attributes)?" Or “What is its degree?”. This progressive questioning enables the construction of a comprehensive diagnostic report based on the findings within the image. Moreover, beyond clinicians, patients and other healthcare workers could also benefit from such a system.
>
> Additionally, while a simple classifier could predict answers, it lacks the interactive element that our VQA model offers. Our model's ability to respond to specific questions adds an extra layer of utility, enabling clinicians to query the system dynamically and receive precise answers relevant to their immediate concerns. This interaction can lead to more personalized and efficient clinical workflows.
>
> Context Processing: In our approach, context extraction serves as a preprocessing step to eliminate ambiguous tokens, including privacy-related replacements. Numeric values are typically transcribed into alphabetic representations, Fig 2, and the measurements we consider pertain to image dimensions and other specifics of the test procedure—not the medical measurements themselves. Importantly, the context does not directly influence model predictions; its primary role is aligning visual features. Consequently, our model exhibits insensitivity to minor variations in context language. We posit that minor modifications to non-medical language would have negligible impact on the extracted visual features.
>
> Dataset and Evaluation: Level1 questions are yes/no questions, while level 2 and 3 can vary. Notably, the attributes in level 3 question refer to the description of the abnormality, example answers include ‘round’ and ‘patchy’. For a comprehensive understanding of the questions and evaluation metrics, please check the original dataset paper [21]. Subsequently, please refer to the dataset GitHub repository for more details on the discrepancy between the released dataset and the results originally published in the paper [21].
>
> We thank the reviewer for highlighting the medical errors in Fig 1 and we are committed to revising and addressing the concerns raised to improve the paper. While acknowledging the existing areas for improvement in our work, we maintain that it aligns with the objectives of the workshop. We eagerly anticipate engaging in profound discussions to delve into its details.

---

### Official Review · Reviewer_JoUH · 2024-07-05

**Recommendation:** 4
**Confidence:** 3

**Clarity:**

The paper is clear and well-written, with minor areas for improvement in clarity

**Feedback:**

- Table 1: The difference in F1 scores is marginal for most of the methods under comparison. Maybe state significance of results, or underline second- and third-best results. Same for recall. Accuracy and Precision speak another language tho.
- Table 2: Left side (hi-VQA) should have bold font for numbers that are better than Context-VQA, e.g. Recall L1.

**Justification:**

The paper presents an interesting idea, with clear benefits regarding the model size (on GPU during training) and training time. However, there are major shortcomings in the evaluation. I still think that this paper is relevant for the workshop, hence I rate it weak accept.

**Reproducibility:**

Sufficient amount of details available for reproducing the main results, but open access is not provided to source code and/or data

**Strengths:**

- The paper is well-organized and well-written.
- The evaluation is quite complete.
- Practical utility is definitely given, and the trade-off between full and partial training are investigated.
- The proposed method significantly beats SOTA, and qualitative results are presented.

**Summary:**

The authors propose a pipeline for Visual Question Answering on chest X-ray imaging, that comprises several steps to extract image context from reports and align it with image features. The proposed method can rely on pre-trained/finetuned models, significantly beating SOTA on the same task.

**Weaknesses:**

- If I understand correctly, the GPT API is used to summarize the original report ("Context Extraction"). In a realistic setting, sensitive data would be leaked, although the entire purpose of this step is to avoid data leakage. Perhaps this step was delegated to GPT 3.5 to facilitate the prototyping, but the manuscript should discuss whether a local model could also be employed for this task.
- Since the pipeline is still quite complex, not only the training time but also the inference time (and FLOPS) matters and should be reported. How fast is the inference of a single test sample (wall clock time)?
- Apparently, the authors created a fully trained and partially trained version of their model (Table 3, Section "Computational Complexity"). However, it is not 100% clear which of these versions are investigated in the ablation study or the baseline comparison, although I strongly assume it is the partially trained (frozen encoders) as this is a central contribution of the paper. It would still be interesting to include both fully and partially trained architectures in all evaluations; I ask myself: "what would I gain with a full training, what do I sacrifice by relying on pre-trained encoders?".
- Source code is not provided.

---

> ### Author Response · Authors · 2024-07-14
> **Rebuttal by Authors**
>
> We thank the reviewer for the thoughtful and constructive feedback on our work. We appreciate your recognition of the strengths in our organization, clarity, and comprehensive evaluation. We also value the suggestions for improving our paper.
>
> Private Data Handling: Our model is trained on public data, which eliminates the risk of privacy exposure. In case sensitive data is exclusively available, a simple preprocessing of data would be sufficient to anonymize personal information such as names and dates before utilizing GPT for further processing. Moreover, Open AI claims following strict privacy and data security protocols for data processed through the GPT API, ensuring that they do not collect or utilize this data to train their language models.
>
> Inference Time: Considering real settings, evaluation on a CPU yields an average inference time of 15 minutes for the entire test set of Rad-Restruct dataset, which is equivalent to an average of 2 seconds per image, including all the questions. This demonstrates the necessary efficiency in handling images with varying numbers of questions. On a GPU, the evaluation is two times faster with approximately 8 min on average. We agree with the reviewer that the inference time is important as well, and commit to including more details, such as FLOPS, during the revision of this paper for the final submission.
>
> Clarity on Model Versions: As the reviewer noted, we reported the results of the partially trained model with frozen encoders. The encoders were robustly pre-trained on large domain specific data, and retraining increases the risk of overfitting on our small dataset. By freezing the encoders, we mitigate this risk and preserve the strong representations already encoded during pre-training. Moreover, retraining the encoders results in training the full architecture which takes time and is computationally costly. Additionally, the partially trained model achieves strong performance, while significantly reducing computational time and complexity, and reducing the number of trainable parameters by 90%.
>
> We recognize that there are areas for improvement in our work, which is one of the reasons we are submitting it to the workshop. Our goal is to engage with professionals to receive valuable insights and feedback that can contribute to refining and advancing our research.

---

### Official Review · Reviewer_eE38 · 2024-07-10

**Recommendation:** 4
**Confidence:** 4

**Clarity:**

The paper is clear and well-written, with minor areas for improvement in clarity

**Feedback:**

I would advise the authors to test a few published medical VLM models on the dataset, and not just the one proposed by Pellegrini et al. This would significantly increase the methodological impact, even if the models were not specifically designed for this task (multi-level complexity of the questions). Furthermore, a convincing motivation for such a VQA setting in a clinical context would be benificial.

**Justification:**

Although there are still some points to improve in the paper, I think it fits well within the scope of the workshop, which aims to provide a framework for in-depth discussions to further enhance the work.

**Reproducibility:**

Sufficient amount of details available for reproducing the main results, but open access is not provided to source code and/or data

**Strengths:**

The paper is well-structured and easy to understand. The methodology and pipeline are thoroughly described and comprehensible. The authors provide sufficient information to estimate the computational complexity of the task. The proposed method shows improvements compared to previous methods on the  Rad-Restruct dataset. Additionally, the authors provide ablation studies to estimate the impact of the proposed module.

**Summary:**

The paper proposes to include medical reports to train an alignment module to generate context-aligned image features. Combined with pretrained Image and Text encoder, this module improves the performance compared to previous methods on the Rad-Restruct dataset..

**Weaknesses:**

Major:
1. The number of baseline methods is a bit sparse. Advanced medical VLMs such as RadFM are missing.
2. The paper mentions that during inference, the model does not receive context information. Please provide a more detailed description of how the module handles this missing information.
3. To enhance reproducibility, it would be beneficial to publish the code.

Minor:
1. Page 2:  “Moreover, we use a pre-trained visual language encoder as an image encoder” – I guess, the authors mean that a vision encoder of a pretrained vision language model is used?
2. Page 2: “By doing so, we significantly reduce the number of (*trainable*)parameters and the computational complexity of our model.”
3. Table 4: Include hi-VQA (Its annoying to switch to Table 1), and add information about which version of hi-VQA was used.

---

> ### Author Response · Authors · 2024-07-14
> **Rebuttal by Authors**
>
> We thank the reviewer for the detailed and constructive feedback. We appreciate their recognition of the strengths in our methodology, pipeline description, and the improvements shown in our results. We are also grateful for the refinement suggestions, which will undoubtedly help us enhance our work. We would like to address some major concerns.
>
> Baseline Methods: Due to time constraints and our commitment to ensuring fair comparisons, the number of baseline methods included in this study is limited. We plan to extend our experiments to incorporate additional methods, such as advanced medical Visual Language Models (VLMs), to provide a more comprehensive comparison.
>
> Inference Context Handling: The attention in f_align’ is trained to learn correlations between the image and context, thereby enabling the extraction of visual features that correspond closely to the context. During the inference stage, in the absence of context, f_align utilizes the learned correlations to align the image feature vectors. Technically, during inference, the trained self-attention mechanism treats the image features as both the value and the query and recalls the learned representations to navigate the image and extract the vision-context vector.
>
> Clinical Application: Our model enables sequential question answering based on an X-ray image. Assuming our method is trained on a large enough dataset, when deployed in clinical settings as a VQA system, users would only need the image and relevant questions without the need of context. In a straightforward scenario, a radiologist may initially inquire, "Is there an abnormality in the image?" The system responds affirmatively and supports subsequent detailed inquiries such as "What is the nature of the abnormality?", "What are the specific signs observed?", "Where is the abnormality located?", and "What are its characteristics (attributes)?" Or “What is its degree?”. This progressive questioning enables the construction of a comprehensive diagnostic report based on the findings within the image. Moreover, beyond clinicians, patients could also benefit from such a system.
>
> The comments will be considered during the refinement of this paper. We hope our clarifications and revisions will address the concerns mentioned and highlight the significance of our contributions and we note that we are considering releasing the source code.
> We are encouraged by the reviewer’s positive recommendation and confidence in our work. We believe addressing these points will greatly improve this paper's quality and impact, and we look forward to having this work deeply discussed during the workshop.

---

### Official Review · Reviewer_cpYe · 2024-07-11

**Recommendation:** 4
**Confidence:** 4

**Clarity:**

The paper is clear and well-written, with minor areas for improvement in clarity

**Feedback:**

I think addressing the questions presented in the weaknesses section will strengthen this paper for journal submission.

Some questions/comments that could be considered for making this study extensive are:

Insights into how the history vector is maintained are needed. Are the questions always presented hierarchically to the system? If evaluating this method on a random image-question pair, how the history is maintained in the absence of the high-level questions at inference time?
What is the impact of context length on the model's performance?

**Justification:**

The proposed work is interesting for the audience and has its merits. The paper is well-written with good presentation. However, there are some questions related to the method and experimental settings (see weaknesses) about this work, addressing them will help understanding this work more clearly.

**Reproducibility:**

Sufficient amount of details available for reproducing the main results, but open access is not provided to source code and/or data

**Strengths:**

1. A well-written and easy-to-follow paper.
2. A new VQA method that leverages clinical reports as context to solve the hierarchical VQA task.
3. Experimental setup is appropriate with good results.

**Summary:**

The paper proposes to solve the VQA task by employing radiology reports as context to image-question pairs during the training phase. The VQA task is formulated as a close-vocabulary answer classification task. A new VQA architecture is introduced that learns to align context with image features to learn better image features by mapping them into the same feature space as the question. The proposed approach is evaluated on Rad-Restruct dataset demonstrating better performance compared to the baseline method hi-VQA notably in terms of the report accuracy, and precision.

**Weaknesses:**

I would like the authors to address the following questions:

1. How the f_align module handles the absence of context at inference needs clarification. How the (context) query is initialized at inference time in this module? An illustration of the f_align module during training and inference will be helpful.
2. This is unclear if open access to the model will be provided or not.
3. The impact of history vector H_i on the final model is not reported in Table 4. It is unclear to me how much beneficial keeping this history vector is. What is the dependency between H_i and a single image-question pair?
4. How often does the context truncation to 512 tokens happen in this dataset? What is the maximum length of the summarized reports?
5. The failure analysis for the proposed Context-VQA method is not provided. It will help in identifying the limitations of this work that require further exploration.
6. The authors claim that Rad-Restruct is the only available VQA dataset with free-text reports. I am aware that the PMC-VQA[1] dataset has QA pairs generated from free text reports. I would like the authors to comment on why did they not consider evaluating their method on this dataset in their experimental setup. What are the limitations?

[1]  PMC-VQA: Visual Instruction Tuning for Medical Visual Question Answering, X. Zhang et al. (https://github.com/xiaoman-zhang/PMC-VQA)

---

> ### Author Response · Authors · 2024-07-14
> **Rebuttal by Authors**
>
> We thank the reviewer for the constructive feedback. We appreciate the positive remarks on the clarity, novelty, and experimental setup of our work. We also value the detailed suggestions for improvement and would like to address the major points raised.
>
> Inference Details:  f_align’s attention is trained to learn correlations between the image and context, thereby enabling the extraction of visual features that correspond closely to the context. During the inference stage, in the absence of context, f_align utilizes the learned correlations to align the image feature vectors. Technically, during inference, the trained self-attention mechanism treats the image as both the value and the query and recalls the learned representations to navigate the image and extract the vision-context vector.
>
> The History Vector, Contex truncation and Failure Cases: The history vector is a list of tokens of previous questions and their answers. For a given image, the history vector is empty for the first question and subsequently updates after each step by concatenating tokens from the preceding question and its answer. Thereby, a single image-question pair is treated similar to the first question with an empty history vector. This autoregressive parsing mechanism enables the model to maintain continuity with previous knowledge (answers) and avoid contradictions in its responses. We are planning additional ablation studies to evaluate the impact of this history vector within our experimental framework. As for the context, most of the reports contain fewer tokens, therefore the truncation rarely occurs on less than 1% of the context during training.
>
> In Figure 3, we present examples of successful cases where our model surpasses previous methods, demonstrating how the quantitative results translate into improved predictions. This illustration aims to provide a clear understanding of the practical implications of our results. It is important to note that our model does encounter failures in other instances, and we are prepared to provide examples of such cases upon request.
>
> The PMC-VQA dataset: Our core motivation is to explore the impact of context on VQA performance. Our choice of close-ended classification on Rad-Restruct dataset was deliberate, as it limits the number of variables that could influence the performance. This task enables experimental validation of our method’s efficacy. The PMC-VQA dataset is designed for generative tasks with significant reliance on text decoders, especially Large Language Models (LLMs) in recent methods. This requires changing the model architecture and complicates attributing improvements to context alone. We are currently in the process of adapting our proposed method to generative VQA. We aim to expand this work and anticipate achieving promising results in the near future. Additionally, we are considering making our source code open to facilitate further research and collaboration.
>
> While we recognize that our work has potential for further refinement, we are eager to engage in discussions during the workshop. We hope these interactions will provide valuable insights and feedback that will help us improve and advance our approach.

---

### Meta-Review · Area_Chair_Jaf2 · 2024-07-16

**Recommendation:** Accept (Oral)
**Confidence:** 4

**Metareview:**

This work presents a technically sound VQA approach using radiology reports as context with good results. The reviewers agree that the paper is clear and well-written.The authors have reasonably addressed the reviewer’s concerns and incorporating these changes will strengthen the paper. Additionally, considering the limitations of data for real-world use would be beneficial as it might not be what doctors actually need. The problem is bigger than this one study, and for this to be useful in a clinical setting, it needs more work. The authors are recommended to include more stronger baseline methods and report statistical significance for a more robust evaluation.

---

### Decision · Program_Chairs · 2024-07-16

Accept (Oral)